# Bone Health, Body Composition, and Vitamin D Status of Black Preadolescent Children in South Africa

**DOI:** 10.3390/nu11061243

**Published:** 2019-05-31

**Authors:** Zelda White, Samantha White, Tasneem Dalvie, Marlena C. Kruger, Amanda Van Zyl, Piet Becker

**Affiliations:** 1Department Human Nutrition, Faculty of Health Sciences, University of Pretoria, Private Bag X20, Hatfield 0028, South Africa; w.samantha297@gmail.com (S.W.); tasneemd0@gmail.com (T.D.); amandajvr@mweb.co.za (A.V.Z.); 2School of Health Sciences, Massey University, Private Bag 11029, Palmerston North 4442, New Zealand; m.c.kruger@massey.ac.nz; 3Faculty of Health Sciences, University of Pretoria, Private Bag X20, Hatfield 0028, South Africa; piet.becker@up.ac.za

**Keywords:** bone health, body composition, vitamin D, preadolescent children, South Africa

## Abstract

Optimal bone health is important in children to reduce the risk of osteoporosis later in life. Both body composition and vitamin D play an important role in bone health. This study aimed to describe bone health, body composition, and vitamin D status, and the relationship between these among a group of conveniently sampled black preadolescent South African children (*n* = 84) using a cross-sectional study. Body composition, bone mineral density (BMD), and bone mineral content (BMC) were assessed using dual x-ray absorptiometry. Levels of 25-hydroxyvitamin D (25(OH)D) (*n* = 59) were assessed using dried blood spots. A quarter (25%) of children presented with low bone mass density for their chronological age (BMD Z-score < −2) and 7% with low BMC-for-age (BMC Z-score < −2), while only 34% of the children had sufficient vitamin D status (25(OH)D ≥ 30 ng/mL). Lean mass was the greatest body compositional determinant for variances observed in bone health measures. Body composition and bone health parameters were not significantly different across vitamin D status groups (*p* > 0.05), except for lumbar spine bone mineral apparent density (LS-BMAD) (*p* < 0.01). No association was found between bone parameters at all sites and levels of 25(OH)D (*p* > 0.05). Further research, using larger representative samples of South African children including all race groups is needed before any conclusions and subsequent recommendation among this population group can be made.

## 1. Introduction

Lifestyle factors (e.g., diet) and body composition affect bone development during growth. In children, lean body mass is one of the strongest correlates of bone mass and bone density, with lean mass (LM) and bone mass having a strong genetic component. The relationship between body composition and bone health is further complicated by the source of adipose tissue (visceral versus subcutaneous), which in turn have different metabolic effects on bone. Bone mass and density are also influenced by other nonmodifiable factors such population ancestry, sex, and maturation [1].

A recent systematic review describing the current evidence on the associations between body composition and bone health in children and adolescents reported consensus that the contribution of LM to the variance of the different bone parameters is larger than the contribution of fat mass (FM) and that an increase in LM is associated with an increase in bone parameters. Contradictory results on the association between body fat and bone parameters were found, being dependent on age and sex. Further research is therefore warranted [2].

The vitamin D status of populations worldwide has been reviewed extensively over the last decade indicating that vitamin D deficiency is re-emerging as a global public health problem in all age groups, with a distinct lack of data in infants, children, and adolescents worldwide, especially in Central and South American and African countries [3,4,5,6]. South African data on vitamin D status among children is limited, with only six published studies, of which four were conducted more than 30 years ago. This limits the relevance of the data to the current vitamin D status of South African children, mainly due to urbanization and the comparability of vitamin D assays used [7].

Vitamin D status has been found to be a key determinant of bone mineral density (BMD) and bone mineral content (BMC) among school children [8,9,10], with greater bone accrual velocity associated with higher parathyroid hormone (PTH) and serum 25-hydroxyvitamin D (25(OH)D) levels in children and adolescents [11]. However, research on the association of vitamin D with bone health outcomes in children is limited and there is a need for further studies in children, especially in diverse racial or ethnic groups [12].

Research studies have observed a relationship between vitamin D status and adiposity in the general population [13], and among children [14,15,16,17,18,19,20]. Clarifying the influence of body weight and body composition on serum 25(OH)D levels has been identified as one of the key knowledge gaps/research needs that need to be addressed in the area of vitamin D nutrition and public health in order to assist in the process of establishing requirements for vitamin D [21], especially considering the increased prevalence of childhood obesity and vitamin D’s key function in bone development.

The aim of this study was to describe the bone health, body composition, and vitamin D status, and the relationship between these among a group of black preadolescent South African children.

## 2. Materials and Methods

### 2.1. Participants

Data were collected by means of a cross-sectional study on 84 conveniently sampled black preadolescent South African children (44 girls, 40 boys; mean ± SD age 8.5 ± 1.4 years) from September to November (spring season) 2016 in Pretoria, South Africa, at a latitude of 25 °S. Ethical approval was obtained from the Research Ethics Committee, Faculty of Health Sciences, University of Pretoria (No: 73/2016). Children from one primary school attending two aftercare facilities in close vicinity to the Prinshof campus of the University of Pretoria (Pretoria, South Africa) were recruited for participation. Informed consent from parents and assent from participants were obtained before data collection.

### 2.2. Anthropometrics

Standing height was measured (to the nearest 0.1 cm) using standard procedures by a single trained researcher with the digital, wireless Seca 274 stadiometer and integrated with body weight assessment (to the nearest 100 g) using the Seca mBCA 514 (version 1.4.292.4928). The anthropometric status of participants was described using height-for-age (HAZ) and body mass index-for-age z-scores (BMI zsc) calculated by the built-in software from the Seca mBCA using the World Health Organization (WHO) reference values [22]. The nutritional status of participants was reported using the categorization used in the South African Demographic and Health Survey [23].

### 2.3. Body Composition and Bone Health

Body composition (fat free mass (FFM), FM and body fat percentage (BF%)) and bone health parameters (BMD, BMC and bone area) were measured by means of whole-body dual X-ray absorptiometry (DXA), using the Hologic Discovery W densitometer (Hologic Inc., Bedford, MA, USA).

Weight estimated by the DXA was used for adjusting bone measurements. The total body less head (TBLH) and lumbar spine (LS) BMD and BMC were used in reporting the data due to the highly reproducible nature of these measures in pediatric measurements [24]. BMD is a 2-dimensional interpretation (g/cm^2^) from the areal measurement. BMD measurements were converted to bone mineral apparent density (BMAD), a volumetric adjustment to areal BMD (aBMD), for greater accuracy by eliminating aBMD errors. BMAD was calculated using the following equations [25,26]:
$$\mathrm{LS}-\mathrm{BMAD}=\frac{\mathrm{BMC}}{{\mathrm{Bone}\;\mathrm{Area}}^{1.5}}$$
$$\mathrm{TBLH}-\mathrm{BMAD}=\frac{\left(\mathrm{BMC}\right)}{\left(\frac{{\mathrm{Bone}\;\mathrm{Area}}^{2}}{\mathrm{height}}\right)}$$

For TBLH bone mass, the Z-score is recommended to be adjusted by the height Z-score also to prevent size-related errors [24]. Z-scores were calculated for both BMD and BMC as indicators for bone health status. The official pediatric position of the International Society for Densitometry (ISCD) states that Z-score values below −2 standard deviations from the mean should be construed as low bone mass for chronological age [27]. Z-scores adjusted for age, gender, and ethnicity were calculated using the following equation [28]:
$$A:\;\mathrm{Bone}\;\mathrm{mass}\;Z-\mathrm{score}=\frac{{\left(\frac{X}{M}\right)}^{L}-1}{\mathrm{LS}}$$
Where variables specific to age, gender, and ethnicity created using LMS curves by Kalkwarf et al. included [29]:

X = DXA BMD/BMC measurement

L = Power in the Box-Cox transformation

M = Median (given as a function of different age percentiles)

S = Standard deviation

The importance of stature was considered and the bone mass Z-score (A) was then adjusted using the height for age z-score (HAZ). The height-adjusted bone mass prediction equation was determined by Zemel et al. using revised reference curves [28]:
B:HAZ bone mass prediction equation= intercept+(HAZ×β)
C:HAZ adjusted Z−score= Bone mass Z (A)−HAZ predicted bone mass (B)
Where: L; M; S; intercept and β = for specific age, ethnicity and gender obtained from Zemel et al. [28].

### 2.4. Vitamin D

Vitamin D status was measured as concentrations of 25(OH)D2 and 25-hydroxyvitamin D3 (25(OH)D3). Data collection took place over 3 days during November 2016. Finger pricks were performed using OneTouch^®^ lancing devices (LifeScan Inc, USA) and blood transferred onto the blood spot cards. The spot cards were dried, sealed, labelled, and sent to ZRT Laboratory (Beaverton, Oregon, USA) for analysis. The 25(OH)D2 and 25(OH)D3 concentrations were determined according to the methodology of Newman et al. [30] by ZRT Laboratory (Beaverton, OR, USA). This blood spot assay has good precision, with interassay coefficients of variation of 11–13% at concentrations of 14, 26, and 81 ng/mL for 25(OH)D3 and 12% at 23 ng/mL for 25(OH)D2. Blood spot 25(OH)D2 and 25(OH)D3 values have also shown very good correlation with serum (r = 0.9) [30]. The 25(OH)D2 and 25(OH)D3 levels were combined and expressed as total 25(OH)D in ng/mL. The Endocrine Society classification was used to classify participants as vitamin D deficient (25(OH)D ≤ 20 ng/mL), insufficient (21–29 ng/mL), or sufficient (≥30 ng/mL) [31].

### 2.5. Statistical Analysis

Data were analyzed using STATA version 14.2 software. The student’s two sample *t*-test was used to compare participant characteristics (anthropometric, Vitamin D status) among gender groups and body composition parameters among vitamin D status groups. To compare bone health parameters of the children categorized by vitamin D status, one-way ANOVA was applied. Tukey’s post-hoc tests were applied where significant differences using one-way ANOVA were found. The mean differences were evaluated at the 5% significance level (*p* ≤ 0.05).

Pearson correlations and logistic linear regression were applied to assess the relationship between body composition parameters and vitamin D status. Simple linear regression models were used in defining the relationship between bone health measures and vitamin D. Adjustments of raw bone health measures for height, gender, age, and body composition factors were done using multiple linear regression. Age as well as age^2^ were included in the regression models because of the nonlinear relationship between bone health measures and age.

## 3. Results

There were 91 children initially recruited for this study. Of these, two did not attend the data collection procedures and five participants’ data were not included in the data analysis due to race/ethnicity being different from the homogenous group of black children. A total of 84 participants took part in the study, of which 59 participants agreed to blood spot analyses for the vitamin D status assessment. Participant characteristics are shown in Table 1.

Forty percent of participants were classified as over-nourished (BMI zsc > +1), with a similar distribution between boys and girls. Fat free mass and BF% differed significantly between boys and girls (*p* < 0.05).

The mean 25(OH)D concentration of the total group was 27.3 ± 5.3 ng/mL, classifying them as vitamin D insufficient (Table 1). Vitamin D sufficient, insufficient, and deficient children, had mean 25(OH)D concentrations of 33.4 ± 3.1 ng/mL, 24.7 ± 2.5 ng/mL, and 19.5 ± 1.0 ng/mL respectively. Only one third (34%) of the children had a sufficient vitamin D status, with boys and girls showing a similar distribution among the different vitamin D status categories. No significant differences were found in vitamin D status between boys and girls (*p* = 0.90).

### 3.1. Bone Health and Body Composition

Of all 84 children assessed by DXA, 25% (*n* = 21) had a BMD Z-score below −2 indicating low bone mass density for chronological age and seven percent (*n* = 6) presented with low BMC-for-age (BMC Z-score < −2), both a sign of poor bone health status. Over-nourished children showed significantly greater raw values of LS-BMD, LS-BMAD, TBLH-BMC, TBLH-BMD, and TBLH-BMAD compared to healthy children (*p* > 0.05).

Lean mass (fat free mass) had a strong statistically significant positive association with TBLH bone parameters and weak statistically significant positive associations with LS bone parameters (*p* < 0.05) (Table 2). Lean mass explained 81%, 79%, and 71% of the variations observed in TBLH-BMD, TBLH-BMC, and TBLH-area respectively. The significant associations found between bone parameters and body weight, and bone parameters and FM were moderate to weak.

Because of the marginal or non-significant associations between BF% and bone parameters, BF% was not included in further adjustments. The greater TBLH-BMC of the over-nourished children compared to healthy children remained statistically significant even after adjustments were made for height, gender, and age (*p* < 0.05). When treating body weight as a confounder, the TBLH-BMC increased for healthy and decreased for over-nourished children to a TBLH-BMC that was not statistically significantly different between the two groups (*p* > 0.05). This was consistent for adjustments made for LM, FM, and LM and FM combined (*p* > 0.05). The raw unadjusted mean values as well as the adjusted LS-BMC mean values did not differ significantly between the healthy and over-nourished children (*p* > 0.05).

### 3.2. Bone Health and Vitamin D Status

Bone health parameters at the LS site and for TBLH between the vitamin D status categories are given in Table 3. The LS-BMAD, calculated to reflect volumetric BMD, was significantly different between the vitamin D sufficient and insufficient, as well as vitamin D sufficient and deficient children of varying vitamin D statuses (*p* < 0.01). Using a Tukey’s post-hoc test, it indicated that the vitamin D insufficient group had a significantly higher LS-BMAD than the deficient group (*p* = 0.003) but not significantly higher than the sufficient group’s LS-BMAD (*p* = 0.098). The vitamin D deficient and sufficient groups did not differ significantly in terms of LS-BMAD (*p* = 0.924). However, no significant difference was observed for the TBLH-BMAD (*p* = 0.35).

Linear regression models further revealed that there were no associations found between all bone parameters at all sites measured and levels of 25(OH)D. The TBLH-BMC, TBLH-BMD, TBLH-area, LS-BMC, and LS-area values were log transformed to better fit the regression model and even under this robust option, the outcome remained the same.

### 3.3. Vitamin D Status and Body Composition

For the purpose of statistical analysis, the vitamin D deficient (*n* = 4) and insufficient (*n* = 35) groups were joined into one group termed vitamin D insufficient (≤29 ng/mL). The 25(OH)D concentrations did not differ significantly between the over-nourished and healthy children (*p* = 0.89). Body composition parameters also did not differ between the vitamin D sufficient and insufficient group (Table 4). Correlations between 25(OH)D concentration and weight (r = −0.04, *p* = 0.78), height (r = 0.08, *p* = 0.55), BMIz (r = −0.07, *p* = 0.62), FM (r = −0.06, *p* = 0.66), FFM (r = 0.04, *p* = 0.76), and BF% (r = −0.08, *p* = 0.55) were non-significant.

Logistic regression analysis with BMI Z-scores and adjusted for BF% indicated that there was a trend towards a two-fold increase in risk for being vitamin D insufficient in obese compared to normal and overweight participants (adjusted OR = 2.6 CI 95%: 0.2; 32.4, *p* = 0.45). Results also showed that boys had a 5% lower risk of being vitamin D insufficient compared to girls; however, the risk was non-significant (adjusted OR = 0.5 CI 95%: 0.13; 2.17, *p* = 0.37).

## 4. Discussion

In this study, we investigated bone health, body composition, and vitamin D status and the relationship between these in a group of black South African preadolescent children.

The total group of children in our study had healthy BMC and BMD for chronological age. Body composition (LM, body weight, and FM) was positively associated with bone health parameters, with LM showing the strongest associations. Other studies on preadolescent children have also found positive associations between FM and volumetric density [32] and bone mass and size [33]. In contrast, negative associations have been found between FM and bone density [34,35]. More recently, being underweight or obese, and having an unhealthy lean or fat mass was reported as predisposing factors for poor bone health among children and adolescents [36].

The FM in this study, had a weaker positive relationship with all bone health measures than LM at all sites. The weaker association of FM on measures of bone mass may be because static loads exerted by FM do not stimulate bone cells involved in the remodeling process as effectively as dynamic loads as with LM [37]. This suggests that the greater bone mass measures observed in over-nourished children is largely attributed to the greater LM and is less likely due to the higher overall FM. A longitudinal study involving overweight and healthy weight children aged 9-11 indicated that changes in bone strength over 16 months calculated using a strength index were associated with changes in LM and not FM [38]. Body fat percentage had weak associations or no association at all with bone mass and bone area. This could be because BF% does not reflect the load bearing on the bones as it does not adjust for the body size [39].

The only significant observation in bone health parameters was for LS-BMAD, which was significantly higher in the vitamin D insufficient group compared to the deficient group, but not significantly higher than the sufficient group’s LS-BMAD. Previous studies have indicated that a higher vitamin D status was positively associated with BMC and aBMD of forearm and whole body in preschool-aged children [10], and positively correlated with BMC and TB-BMD in prepubertal nonobese children [9]. Serum 25(OH)D concentrations was also identified as an independent determinant of lumbar spine and whole body BMD among Finnish children and adolescents (age range 7–19 years) [8]. A three-year prospective study examined the association between changes in BMD or BMAD and serum 25(OH)D in 171 healthy Finnish girls aged 9–15 years. Results indicated that baseline 25(OH)D correlated significantly with the unadjusted three-year change in BMD at the lumbar spine and femoral neck in all participants. They concluded that pubertal girls with hypovitaminosis D seem to be at risk of not reaching maximum peak bone mass, particularly at the lumbar spine [40]. In this study, no correlation was found between 25(OH)D and bone health measures. Similar results were reported by Marwaha et al., who did not find any significant association of either 25(OH)D or PTH with either site in healthy school children in Northern India [41]. Intervention studies with vitamin D supplementation have confirmed this lack of correlation. Although daily supplementation with high-dose cholecalciferol (vitamin D3) in children and young adults with HIV infection for 12 months resulted in increased 25(OH)D concentrations, no impact was found on bone density, structure, and strength [42]. Similarly, PTH and bone turnover markers were not associated with 25(OH)D at baseline or with change in 25(OH)D over 12 weeks in black and white children [43].

The minimum desirable levels of 25(OH)D in children is 20 ng/mL, while a level of 12 ng/mL would be the minimum acceptable on the basis of PTH response and normal levels found in summer months [44]. This minimum level is classified as deficient by the Endocrine Society. In our study, only 7% of the children had 25(OH)D at or below 20 ng/mL, which could explain why differences in bone mass were not found to be associated with vitamin D.

Although adequate serum 25(OH)D levels based on bone mass in children have not yet been confirmed, the relationship between PTH, bone health and 25(OH)D also presents limitations in providing insight to establish a threshold for the vitamin D status biomarker. An inverse relationship between 25(OH)D levels and PTH concentration has been reported for preadolescents and adolescents [45,46] and it has been suggested that the level of 25(OH)D at which PTH is suppressed could provide insight in to what the 25(OH)D threshold for vitamin D deficiency may be. However, the relationship between PTH and vitamin D in bone metabolism is also complicated, since PTH concentrations, like 25(OH)D, depends on ethnicity. Parathyroid hormone has been found to be significantly higher in black populations in comparison to other ethnic groups [47] and thus using this as an indication for a 25(OH)D threshold would provide inaccuracies. Changing concentrations of PTH due to pubertal factors is another reason this biomarker would be a false indicator of the relation of 25(OH)D and bone mass [48,49]. The mean 25(OH)D concentration of 27.3 ng/mL found in our group of children classify them as vitamin D insufficient according to the Endocrine Society classification [31]. However, the Institute of Medicine (IOM) classification of vitamin D insufficiency is much more conservative (12–20 ng/mL). When applying this cut-off to our study participants, the percentage of vitamin D insufficiency decreased from 59% (*n* = 35) to 7% (*n* = 4) [50]. Optimal serum 25(OH)D concentrations are however still being debated with no clear consensus [51].

Two previous South African studies reported a mean serum 25(OH)D concentration of 50 ng/mL (6–9 years) [52] and 34.5 ng/mL (10 years) [18] among healthy black children, both classifying them as vitamin D sufficient. A similar number of black children were classified as vitamin D deficient (8%) in the study by Poopedi et al. compared to our study (7%), although the number of vitamin D sufficient children was double (70%) what was found in our study (34%) [18]. Possible factors reported previously affecting serum 25(OH)D concentrations that could be excluded as reasons for differences observed in our study compared to the two previous studies among black children include: age (similar age ranges covered), sex (studies included boys and girls), latitude (varied between 24–26 degrees south), and winter season (our study and Poopedi et al. [18] obtained data in the spring season). Attributable factors could therefore include reduced sunlight exposure, dietary habits, and possible differences in body composition among groups across the different studies. It is important to note that research has found that blacks are able to maintain a healthy serum 1,25(OH)D2 level despite the fact that they have deficient 25(OH)D levels [47]. It is therefore recommended that that measurement of the serum 1,25(OH)D2 level or the amount of the bioavailable 25(OH)D in the serum would be more accurate to determine vitamin D sufficiency or deficiency in the black population [47].

The 25(OH)D concentrations of our children were not related to any body composition measurement. Poopedi et al., on the other hand, reported significantly negative correlations with percentage of fat tissue and positively correlations with percentage lean tissue among black South African children. A significant correlation was also found between FM and 25(OH)D after adjusting FM and LM for height [18].

The relationship between vitamin D and several obesity and body composition indices have also been reported in other child populations. Vitamin D insufficiency has been associated with increased BMI among obese children residing in Brooklyn (NY) [14], and significantly higher BMI has been reported in vitamin D deficient compared to sufficient groups among Spanish [17] and obese American children from different race groups [16]. Children with insufficient vitamin D also reported higher body weight and waist measurement [17,19], waist/height ratio [17], triceps and subscapular skinfold thickness [19], and FM [16] compared to children with adequate vitamin D. Fat mass was also negatively correlated with 25(OH) D, without seasonal and racial/ethnic influences among obese American children [16].

A very recent study among Canadian youths (6–17 years) confirmed the inverse association between overweight/obesity and serum 25(OH)D for both sexes after adjustment for age, race, income, season, vitamin D supplementation, and daily milk consumption [14]. Using a multiple linear regression analysis, only weight and BMI in Spanish [17] and weight, BMI Z-score, waist circumference, and triceps and subscapular skinfold thickness (as well as the sum of both) in Belgian children (adjusted for age, month of sampling, and hours playing outside per week) [19] were found to independently influence 25(OH)D values.

In our study, we observed a trend towards a two-fold increase risk for being vitamin D insufficient in obese compared to normal and overweight participants. Greene-Finestone et al. also reported that obese or overweight boys had over twice the odds of having 25(OH)D levels <16 ng/mL and <20 ng/mL compared with normal weight boys, while obese or overweight girls had 1.4 times the odds of levels <20 ng/mL [20].

The small sample size included in our study could have affected the strength and significance of associations between 25(OH)D and body composition measurements. Although it is still unclear whether vitamin D deficiency causes obesity or vice versa, a study by Reinehr et al. investigated the relationships between 1,25(OH)D, 25(OH)D, and weight status before and after weight loss in obese children (*n* = 133; mean age 12.1 years). Obese children had significantly lower 25(OH)D concentrations compared with non-obese children, while 1,25(OH)D did not differ significantly. Changes of 25(OH)D correlated significantly with changes in BMI standard deviation score (SDS-BMI). A reduction of overweight in 35 children led to a significant increase in 25(OH)D levels, indicating that these changes are the consequence rather than the cause of overweight [15].

The major limitations of this study include the relative small sample size of children included, as well as the non-representative sample from which the results cannot be generalized to the broader black preadolescent South African population. The assumption that the children were pre-menarche, particularly the 9–10 year old girls, is also a limitation and it is recommended that future studies should include the assessment of pubertal maturation. When investigating the relation between vitamin D and bone health, it is recommended to include serum calcium and PTH assessment, due to the importance of calcium in bone health and the interrelationship of calcium, PTH and serum 25(OH)D. This is especially vital in black populations, due to racial differences found, and the existence of the African paradox of lower fracture risk despite lower serum 25(OH)D [48].

Blood samples obtained by finger pricks are less invasive compared to venous blood draws and were used in this study to prevent a decrease in study participation. 25(OH)D levels determined by mass spectrometry from blood spot cards are on average 1.1 ng/mL lower compared to serum samples with a non-significant 3.3 percent difference (95% CI: 26.3–12.1%; *p* = 0.48) using Bland-Altman plots. Blood spot measurements may therefore not substantially affect the prevalence of insufficiency and deficiency [53].

The strengths of this cross-sectional study lie in the assessments used for vitamin D and bone health. Both DXA and liquid chromatography/tandem mass spectrometry are considered the gold standards for the relative data collections and allow for accurate, precise, and reliable results. This study also contributes to the limited knowledge of vitamin D status among black preadolescent South African children, and its relation to body composition and bone health. It is recommended that further research, using larger representative samples of South African children including all race groups, is needed before any conclusions and subsequent recommendation for vitamin D among this population group can be made.

## Figures and Tables

**Table 1 nutrients-11-01243-t001:** Participant characteristics for total group and comparison between genders.

**Parameters**	**Total (*n* = 84)**	**Boys (*n* = 40)**	**Girls (*n* = 44)**	***p***
Age (years)	8.6 ± 1.4 ^1^	8.6 ± 1.5 ^1^	8.6 ± 1.3 ^1^	0.98
Height (cm)	1.32 ± 0.10 ^1^	1.32 ± 0.10 ^1^	1.32 ± 0.09 ^1^	0.94
Weight (kg)	33.5 ± 10.9 ^1^	33.4 ± 11.1 ^1^	33.7 ± 11.0 ^1^	0.91
HAZ	0.37 ± 0.94 ^1^	0.34 ± 0.95 ^1^	0.40 ± 0.14 ^1^	0.76
BMI zsc	1.02 ± 1.63 ^1^	1.05 ± 1.82 ^1^	0.99 ± 1.47 ^1^	0.87
Healthy (BMI zsc −2–+1)	50 (60%) ^2^	25 (63%) ^2^	25 (57%) ^2^	
Over-nourished (BMI zsc >+1)	34 (40%) ^2^	15 (38%) ^2^	19 (43%) ^2^	
Body composition				
Fat mass (kg)	11.7 ± 7.0 ^1^	10.3 ± 6.9 ^1^	12.9 ± 7.0 ^1^	0.10
Fat free mass (kg)	21.5 ±4.9 ^1^	22.6 ± 5.0 ^1^	20.5 ± 4.7 ^1^	0.04
Body fat (%)	32.9 ± 9.7 ^1^	28.9 ± 9.2 ^1^	36.5 ± 8.8 ^1^	<0.01
**Vitamin D status**	**Total (*n* = 59)**	**Boys (*n* = 28)**	**Girls (*n* = 31)**	***p***
25(OH)D (ng/mL)	27.3 ± 5.3 ^1^	28.4 ± 5.1 ^1^	26.3 ± 5.2 ^1^	0.90
Deficient (≤20 ng/mL)	4 (7%) ^2^	2 (7%) ^2^	2 (6%) ^2^	
Insufficient (21–29 ng/mL)	35 (59%) ^2^	15 (54%) ^2^	20 (65%) ^2^	
Sufficient (≥30 ng/mL)	20 (34%) ^2^	11 (39%) ^2^	9 (29%) ^2^	

^1^ mean ± SD; ^2^ number (column percentage). HAZ: height for age z-score; BMI zsc: body mass index z-score.

**Table 2 nutrients-11-01243-t002:** Association between bone health parameters and body composition.

Bone Health Parameters	Body Weight (kg)	Lean Mass (kg)	Fat Mass (kg)	Body Fat (%)
**TBLH-BMC (g)**	11.25 (1.04)	29.83 (1.58)	13.11 (2.01)	4.76 (1.72)
***p***	<0.001	<0.001	<0.001	0.007
**R^2^**	0.59	0.81	0.34	0.09
**LS-BMC (g)**	0.18 (0.04)	0.48 (0.08)	0.19 (0.06)	0.06 (0.05)
***p***	<0.001	<0.001	0.002	0.162
**R^2^**	0.22	0.32	0.11	0.02
**TBLH-BMD (g/cm^3^)**	0.006 (0.0005)	0.014 (0.0008)	0.007 (0.001)	0.003 (0.0008)
***p***	<0.001	<0.001	<0.001	0.001
**R^2^**	0.59	0.79	0.37	0.12
**LS-BMD (g/cm^2^)**	0.004 (0.0007)	0.01 (0.002)	0.006 (0.001)	0.003 (0.0009)
***p***	<0.001	<0.001	<0.001	<0.001
**R^2^**	0.33	0.32	0.27	0.15
**TBLH-Area (cm^2^)**	8.35 (0.94)	22.68 (1.60)	9.31 (1.74)	2.91 (1.43)
***p***	<0.001	<0.001	<0.001	0.045
**R^2^**	0.49	0.71	0.26	0.05
**LS-Area (cm^2^)**	0.09 (0.04)	0.34 (0.09)	0.04 (0.06)	−0.04 (−0.05)
***p***	0.035	<0.001	0.53	0.44
**R^2^**	0.05	0.16	0.005	0.007

Values reported as β_1_ (SE); TBLH: total body less head; BMC: bone mineral content; LS: lumbar spine; BMD: bone mineral density; BMAD: bone mineral apparent density.

**Table 3 nutrients-11-01243-t003:** Bone health parameters of participants in relation to vitamin D status.

Bone Health Parameters	Total Population (*n* = 59)	Deficient (≤20 ng/mL) (*n* = 4)	Insufficient (21–29 ng/mL) (*n* = 35)	Sufficient (≥30 ng/mL) (*n* = 20)	*p*
**LS-BMC (g)**	18.5 ± 4.0	17.5 ± 3.4	18.4 ± 4.4	18.7 ± 3.5	0.86
**LS-BMD (g/cm^2^)**	0.686 ± 0.080	0.640 ± 0.080	0.704 ± 0.077	0.665 ± 0.077	0.10
**LS-BMAD (g/cm^3^)**	0.133 ± 0.016	0.123 ± 0.013 ^b^	0.139 ± 0.016 ^a^	0.126 ± 0.013 ^ab^	<0.01
**LS-area (cm^2^)**	26.8 ± 3.9	27.3 ± 3.0	26.0 ± 4.2	28.0 ± 3.1	0.18
**TBLH-BMC (g)**	657.0 ± 153.9	623.4 ± 150.5	664.0 ± 160.9	651.3 ± 148.3	0.87
**TBLH-BMD (g/cm^2^)**	0.637 ± 0.076	0.612 ± 0.076	0.643 ± 0.077	0.631 ± 0.075	0.68
**TBLH-BMAD (g/cm^3^)**	0.083 ± 0.008	0.078 ± 0.005	0.084 ± 0.008	0.082 ± 0.008	0.35
**TBLH-area (cm^2^)**	1019.9 ± 127.6	1066.1 ± 137.6	1020.9 ± 134.0	1021.0 ± 120.5	0.98

Values reported as mean ± SD. LS: lumbar spine; TBLH: total body less head; BMC: bone mineral content; BMD: bone mineral density; BMAD: bone mineral apparent density. Means bearing different superscript letters differ significantly.

**Table 4 nutrients-11-01243-t004:** Vitamin D status in relation to body composition parameters (*n* = 59).

Body Composition Parameters	Vitamin D Insufficient (≤29 ng/mL) (*n* = 39)	Vitamin D Sufficient (≥30 ng/mL) (*n* = 20)	*p*
BMI zsc	1.39 ± 1.8 ^1^	0.97 ± 1.4 ^1^	0.43
Lean mass/Fat Free Mass (kg)	22.3 ± 5.1 ^1^	21.3 ± 3.8 ^1^	0.51
Fat mass (kg)	13.0 ± 7.8 ^1^	11.2 ± 5.3 ^1^	0.43
Body fat (%)	34.3 ± 10.4 ^1^	33.1 ± 7.6 ^1^	0.68
Healthy (BMI zsc −2–+1)	19 (49%) ^2^	11 (55%) ^2^	
Over-nourished (BMI zsc > +1)	20 (51%) ^2^	9 (45%) ^2^	

^1^ mean ± SD; ^2^ number (column percentage). BMI zsc: body mass index z-score.

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
