# Peer review of "Bone Health, Body Composition, and Vitamin D Status of Black Preadolescent Children in South Africa"

_nutrients, 2019, doi:10.3390/nu11061243_

Reviewer 1 Report

In this manuscript, the authors showed no association between serum levels of 25(OH)D and bone parameters, assessed by DXA. No association is consistent with the results of the recently published large cohort studies with vitamin D supplements; this information should be highlighted in the Discussion section.  

The extracted data collected from a cross-sectional study is well presented. However, authors need to let the readers know why 35 children with that serum 25(OH)D level of 24.7ng/ml is considered as insufficient. How does the result look like if the cutoff value is considered as 20ng/ml, as recommended by the IOM. Also, considering<30ng/ml as hypovitaminosis D is confusing.

Authors are very correct to suggest that any recommendation of vitamin D supplements need further studies with larger samples. Moreover, supplements are not as safe as perceived (J Steroid Biochem Mol Biol. 2018 Jun; 180:81-86).  

Author Response

Point 1: In this manuscript, the authors showed no association between serum levels of 25(OH)D and bone parameters, assessed by DXA. No association is consistent with the results of the recently published large cohort studies with vitamin D supplements; this information should be highlighted in the Discussion section

 Response 1: The results of the association between serum levels of 25(OH)D and bone parameters have been compared to studies on preadolescent/school children showing similar and opposing results. (Line 239-251):

It is however not clear to the author to which specific “large cohort studies” the reviewer refers to (adult/children), but the authors did include the following recent intervention studies of vitamin D supplementation and bone parametres found in children as part of the discussion as recommended (Line 252-257):

43. Rovner, A.J.; Stallings, V.A.; Rutstein, R.; Schall, J.I.; Leonard, M.B.; Zemel, B.S. Effect of high-dose cholecalciferol (vitamin D3) on bone and body composition in children and young adults with HIV infection: a randomized, double-blind, placebo-controlled trial. Osteoporos Int. 2017, 28:201–209.

43. Rajakumar, K.; Moore, C.G.; Yabes, J.; Olabopo, F.; Haralam, M.A.; Comer, D.; Bogusz J.; Nucci, A.; Sereika, S.; Dunbar-Jacob, J.; Holick, M.F.; Greenspan, S.L. Effect of Vitamin D3 Supplementation in Black and in White Children: A Randomized, Placebo-Controlled Trial. J Clin Endocrinol Metab. 2015, Aug 100(8):3183–3192.

 Point 2: The extracted data collected from a cross-sectional study is well presented. However, authors need to let the readers know why 35 children with that serum 25(OH)D level of 24.7ng/ml is considered as insufficient. How does the result look like if the cutoff value is considered as 20ng/ml, as recommended by the IOM. Also, considering<30ng/ml as hypovitaminosis D is confusing.

 Response 2:

In this study, the authors used the Endocrine Society classification for the classification of participant’s vitamin D status (Methods section: line 129-131). The classification used has now been indicated in the discussion section as well (line 275-276). The authors also added the IOM classification and indicated the number of insufficient participants when using the IOM classification, and also mentioned the debate and lack of consensus on the classification and optimal serum 25(OH)D (line 257-279).

 Line 203-204: “For the purpose of statistical analysis, the vitamin D deficient (n=4) and insufficient (n=35) groups were joined into one group termed Hypovitaminosis D (≤ 29ng/ml).” Based on the confusion expressed by the reviewer, the authors changed the “hypovitaminoses D” to “vitamin D insufficient” throughout the results and discussion section.

 Reviewer 2 Report

It is an interesting study on bone health, body composition and vitamin D status in children.

I think that lack of determination of PTH and calcium is one of the major limitations of this study.

These parameters should be included in the assessment of bone and vitamin status, especially in black children.

Line 121 - it is not possible that data collection took place during November 2019.

Author Response

Point 1: I think that lack of determination of PTH and calcium is one of the major limitations of this study. These parameters should be included in the assessment of bone and vitamin status, especially in black children.

Response 1: The author agree with this comment and has indicated this under the limitations (and subsequent recommendation) of the study. These measurements are however, invasive measurements and could affect recruitment/participation as well as ethical clearance from various authorities.

 Point 2: Line 121 - it is not possible that data collection took place during November 2019.

  Response 2: This was an error. Year has been corrected on manuscript, should be 2016.

Round  2

Reviewer 1 Report

Revised accordingly, I've no additional issue with this manuscript.

Reviewer 2 Report

I think that this is accepted form of your manuscript.